# Thermo-optical bistability in silicon micro-cantilevers

Basile Pottier, Ludovic Bellon[*]

Univ Lyon, Ens de Lyon, CNRS, Laboratoire de Physique, F-69342 Lyon, France
* ludovic.bellon@ens-lyon.fr

April 21, 2021

## Abstract

We report a thermo-optical bistability observed in silicon micro-cantilevers irradiated by a laser beam with mW powers: reflectivity, transmissivity, absorption, and temperature can change by a factor of two between two stable states for the same input power. The temperature dependency of the absorption at the origin of the bistability results from interferences between internal reflections in the cantilever thickness, acting as a lossy Fabry-Pérot cavity. A theoretical model describing the thermo-optical coupling is presented. The experimental results obtained for silicon cantilevers irradiated in vacuum at two different visible wavelengths are in quantitative agreement with the predictions of this model.

# 1 Introduction

Micrometer sized resonators, such as membranes or cantilever, are used as precision sensors in a broad range of applications: mass sensing down to $10^{-18}$ g [1], single molecule light absorption imaging [2], force detection with $10^{-19}$ N sensitivity in Atomic Force Microscopy (AFM) [3], quantum measurements [4], to cite just a few applications. To address those mechanical devices and actually perform the measurement, light is used in many actuation [5–9] and sensing schemes [10–15]. Understanding light-matter interaction in those systems is thus important to fully exploit the devices and discard artifacts from the measurements.

In AFM for example, a cantilever carries a sharp tip that scans a sample and the force of interaction between the two is usually measured by an optical readout (optical beam deflection [10] or interferometry [11–15]). A modulated light power on the cantilever can also be used in AFM to drive it at resonance in dynamic measurements [5–9]. This actuation is a photo-thermal effect: the light absorbed locally heats and bends the cantilever. Indeed, using moderate laser powers (a few mW) on AFM silicon cantilevers, one can reach huge temperatures [16, 17] (up to silicon melting at 1410 °C) owing to the relatively large absorption of visible light by silicon.

In this article, we present and fully characterize an unexpected effect of the light-matter interaction in AFM silicon micro-cantilevers: a thermo-optical bistability. An optically bistable system is one that can exhibit two output states for the same optical input. As illustrated in Fig. 1, such an effect can arise on a standard cantilever irradiated by a red laser beam, with reflectivity changing by a factor 2 between two stable states in a 1.4 mW range around 7 mW of incident light power.

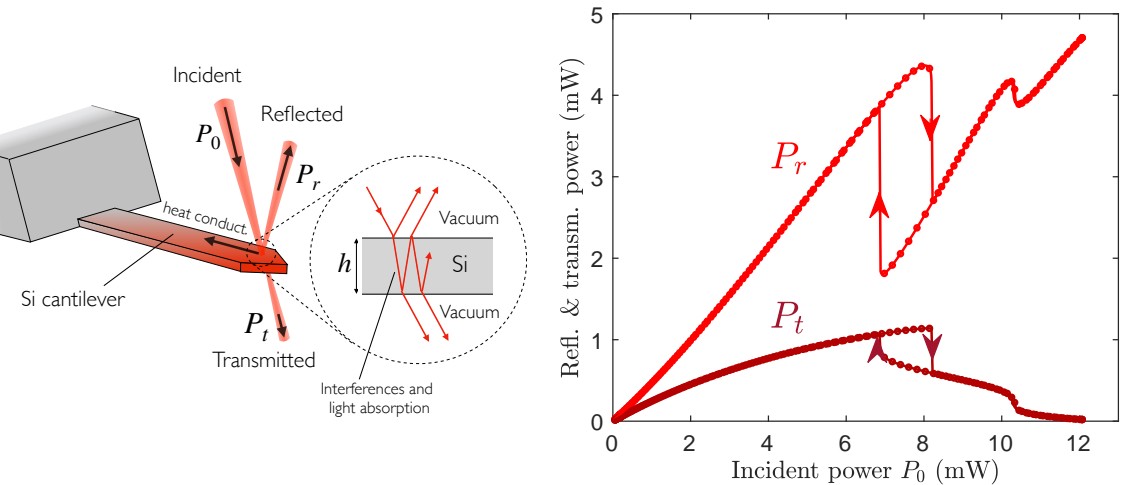

Figure 1: Measured reflected ($P_r$) and transmitted ($P_t$) optical powers as a function of the incident one ($P_0$) of a laser beam (641 nm) focused at the free end of a silicon cantilever (C14, see text) in vacuum. The cantilever thickness acts as a Fabry-Pérot cavity with absorption for visible light. This results in a temperature rise at the tip, driving a thermo-optical coupling, and leading to the measured non-linear characteristics and the strong hysteresis around 7 mW. Light beams are actually perpendicular to the cantilever surface and have been represented at an angle for illustration purposes.

With the long-standing goal to replace the electronic transistor by an all-optical device, optical bistability has been studied extensively on its own for the last 50 years [18–29]. The use of this fundamental phenomenon makes it possible to perform optical memory, amplifier, and logic functions. The simplest examples of working devices are Fabry-Pérot cavities filled

with a material having a non-linear optical response. In refractive bistability, the non-linearity is achieved using a material in which the refractive index is intensity-dependent. This can be achieved either directly with the Kerr effect [20, 21] or indirectly using the temperature dependency of the index and some absorbing material to couple the temperature to the intensity [23, 24, 30–32]. This latter case, often referred to as thermo-optic bistability, applies to our system: we demonstrate in this article that the observed hysteresis is rooted to the non-linear absorption of light in the cantilever thickness, acting as a lossy Fabry-Pérot cavity, coupled to the temperature increase generated by this heat source.

In the following, we introduce in section 2 an overview of the observations and of the mechanism responsible for the bistability. Section 3 presents a comprehensive theoretical framework of the phenomenon. We follow with detailed experimental results on two different cantilevers (section 4), and a quantitative comparison to theory (section 5), before a discussion on the perspectives of this work in conclusion.

## 2 Optical response of a silicon cantilever irradiated in vacuum

For visible light, silicon is neither transparent nor a good mirror (reflectivity of around 37%). Bare silicon cantilevers thus potentially absorb a significant fraction of light. When the cantilever is in vacuum and a laser beam is focused at its free end (Fig. 1), the heat generated by absorption can only dissipate via conduction along cantilever length through the tiny cross-section, resulting in significant temperature rise [16, 17, 33, 34]. In addition, owing to the relatively weak absorption in silicon, the light experiences multiple reflections within cantilever thickness which acts as a Fabry-Pérot cavity. Consequently, the fraction of power absorbed results from the interferences between internal reflections and is expected to be an oscillating function of the round-trip phase difference

$$\phi = \frac{4\pi}{\lambda} nh, \tag{1}$$

where $\lambda$ is the wavelength of light, $n$ is the silicon refractive index and $h$ the cantilever thickness. A rise in the cantilever temperature will lead to a change in absorption through both the temperature dependency of silicon refractive index and the thermal expansion. In other words, the absorbed power, at the origin of the temperature rise, depends itself on temperature through the effect of interferences. Depending on the sign of variation of absorption with temperature, we distinguish two different behaviors. For an absorption increasing with temperature, the thermo-optical coupling will tend to rise further the temperature. Oppositely, for an absorption decreasing with temperature, the thermo-optical coupling has a stabilizing effect.

In Fig. 1, we display both the reflected and transmitted powers measured as a function of the incident one $P_0$ when shining a laser on a raw silicon cantilever in vacuum. For a non-absorbing cantilever or in the case the effect of interference would be negligible, the characteristic would be linear (appearing as straight lines). Here, both characteristics are non-linear and further exhibit a bistability around 7 mW which can be explained by the thermo-optical mechanism introduced above. In the next section, we present a theoretical model to describe quantitatively the bistable behavior observed.

## 3 Theoretical model

### 3.1 Optical absorption of a silicon film

We consider a plane parallel film of thickness $h$ illuminated by a plane wave of monochromatic light upon normal incident radiation. In the general case, the film is absorbing, the optical properties are expressed in terms of a complex refractive index $\tilde{n} = n + i\kappa$. The film can be considered as a symmetric Fabry-Pérot cavity with internal loss. The transmitted $a_t$ and reflected $a_r$ electrical field amplitudes experience multiple reflections between the film surfaces and can be calculated as a geometrical series

$$a_t = a_0 \left( tt'e^{i\beta\tilde{n}h} \sum_{m=0}^{\infty} r'^{2m} e^{2i\beta m\tilde{n}h} \right) \tag{2a}$$

$$a_r = a_0 \left( r + tt'r' \sum_{m=0}^{\infty} r'^{2m} e^{2i\beta(m+1)\tilde{n}h} \right) \tag{2b}$$

where $a_0$ is the incident electrical field amplitude, $\beta = 2\pi/\lambda$ is the wavenumber. $t, r$ denote the transmission and reflection coefficients for a wave traveling from the surrounding medium into the film and $t', r'$ the corresponding coefficients for a wave traveling in the opposite direction. These coefficients are given by the Fresnel formulae. In the case the film is placed in vacuum, we have $r = (1 - \tilde{n})/(1 + \tilde{n})$, $r' = -r$, $t = 2/(1 + \tilde{n})$, $t' = 2\tilde{n}/(1 + \tilde{n})$. Taking the square absolute value of Eqs. (2) leads directly to the corresponding intensities. The power transmission $T$ and power reflection $R$ coefficients read as

$$T = \frac{|a_t|^2}{|a_0|^2} = \frac{\left(1 - 2\mathscr{R}\cos(2\psi) + \mathscr{R}^2\right)e^{-\alpha h}}{(1 - \mathscr{R}e^{-\alpha h})^2 + 4\mathscr{R}e^{-\alpha h}\sin^2(\phi/2 + \psi)}, \tag{3a}$$

$$R = \frac{|a_r|^2}{|a_0|^2} = \frac{\mathscr{R}\left((1 - e^{-\alpha h})^2 + 4e^{-\alpha h}\sin^2(\phi/2)\right)}{(1 - \mathscr{R}e^{-\alpha h})^2 + 4\mathscr{R}e^{-\alpha h}\sin^2(\phi/2 + \psi)}, \tag{3b}$$

where $\alpha^{-1} = \lambda/4\pi\kappa$ is the penetration depth of light, $\psi = \arctan(2\kappa/n^2 + \kappa^2 - 1)$ is the phase shift on internal reflection, and $\mathscr{R} = ((n-1)^2 + \kappa^2)/((n+1)^2 + \kappa^2)$ is the power reflectivity at film surfaces. For a non-absorbing film ($\kappa = 0$), Eqs. (3) reduce to the well-known Airy functions [35].

The fraction $A$ of light power absorbed by the film is obtained as the complementary of transmission and reflectivity as $A = 1 - R - T$. Note that light absorption is non-uniform within the film thickness. The absorption coefficient $A$ calculated here corresponds to the total absorption. Through Eqs. (3), the behavior of $A$ as a function of phase difference $\phi$ strongly depends on $\alpha h$, the relative importance of the intrinsic absorption to the film thickness $h$. For a relatively thin film (or a relatively transparent material), corresponding to $\alpha h \ll 1$, at first order the film absorption is negligible. On the opposite, for a relatively thick film $\alpha h \gg 1$, the interferences are negligible, the film absorption remains constant, $A = 1 - \mathscr{R}$. For silicon, the intrinsic absorption length strongly increases with wavelength. As a result, a $\mu$m silicon film may be considered opaque in ultraviolet and transparent in infrared (Fig. 2). In visible light, the silicon film is semi-transparent, $A$ is relatively important and exhibits strong oscillations as the phase difference $\phi$ varies. For a given wavelength $\lambda$, the separation of adjacent local maxima (or minima) in $A$ corresponds to a change of $\lambda/2n$ in thickness. It corresponds to 65 nm at $\lambda = 532$ nm and 84 nm at $\lambda = 641$ nm.

The silicon film thus acts as a lossy Fabry-Pérot cavity. Due to the high absorption, it is an intrinsically bad resonator: at $\lambda = 641$ nm for $h = 2\,\mu$m, the wavelength full-width at half-maximum of the transmission oscillation amplitude is $\delta\lambda \sim 16$ nm, while the free spectral

range is $\Delta\lambda \sim 21$ nm, leading to a finesse $\mathscr{F} = \Delta\lambda/\delta\lambda \sim 1.3$, and a quality factor of the optical resonator $\mathscr{Q} = \lambda/\delta\lambda \sim 40$. Better optical figures of merit could be reach by tuning the wavelength towards infrared.

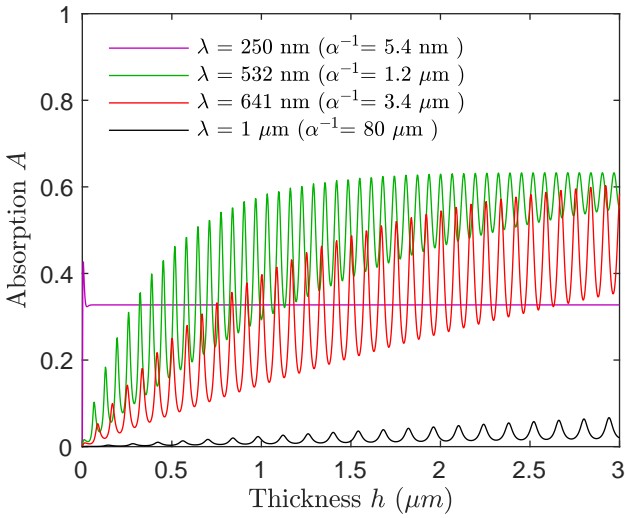

Figure 2: Absorption coefficient $A$ of a silicon film as a function of its thickness $h$ for various wavelengths (using the refractive index at 300 K from Ref. [36]). In visible light, the penetration depth of light is comparable to the film thickness $\alpha h \approx 1$, the film absorption is relatively important and exhibits large oscillations caused by the interferences.

## 3.2 Temperature dependency of optical absorption

A change in the film temperature leads to a shift of the phase difference due to both, a change in the refractive index $n$ as well as the film thickness $h$ via thermal expansion. For silicon in the visible spectral range, $n$ and $\kappa$ from room temperature up to around 500 °C can be respectively parameterized as a linear and an exponential function of temperature [37–39]:

$$n(\theta) = n_0(1 + a_n\theta), \tag{4a}$$
$$\kappa(\theta) = \kappa_0\exp(\theta/\theta^\star), \tag{4b}$$

where the parameters $n_0$, $a_n$, $\kappa_0$ and $\theta^\star$ depend on the considered wavelength. The cantilever thickness $h$ varies also with temperature due to dilatation as

$$h(\theta) = H(1 + a_h\theta), \tag{5}$$

where $a_h$ is the silicon thermal expansion coefficient and $H$ the cantilever thickness at room temperature. For silicon, $a_h$ depends on temperature and can be described from room temperature up to 1500 K by an empirical formula given by Ref. [40].

The phase difference shift $\Delta\phi$ induced by a temperature difference $\Delta\theta$ reads as

$$\Delta\phi = \frac{4\pi}{\lambda}n_0H(a_n + a_h)\Delta\theta. \tag{6}$$

For silicon, $a_n = 98 \times 10^{-6}\,\mathrm{K}^{-1}$ (at $\lambda = 641$ nm [37]) and $a_h = 2.6 \times 10^{-6}\,\mathrm{K}^{-1}$ [40] at room temperature. The dominant mechanism responsible for the phase shift $\Delta\phi$ is therefore the change in the refractive index. For a film such that $\alpha h \approx 1$, this phase shift may trigger significant absorption variations. The film absorption $A$ will turn from a minimum to a maximum

value for $\Delta\phi = \pi$, corresponding to a temperature change $\Delta\theta = \lambda/[4Hn_0(a_n + a_h)]$. For silicon at $\lambda = 641$ nm, we get $\Delta\theta = 159$ K with $h = 2.6\,\mu$m.

In Eq. (6), we notice that the optical phase driving the interference state inside the cavity is proportional to $H$ and $\theta$ and inversely proportional to $\lambda$. There is thus a direct link between the thermo-optical coupling stabilising/destabilising effect (sign of $dA/d\theta$) and where the wavelength stands with respect to the Fabry-Pérot resonance. Indeed, if $\lambda$ is above the resonance, increasing $\theta$ is equivalent to decreasing $\lambda$, thus getting closer to the resonance and increasing $A$: $dA/d\theta > 0$. On the other side of the resonance, we get on the contrary $dA/d\theta < 0$.

As described by Eq. (4b), the extinction coefficient $\kappa$ varies with temperature. For silicon in visible light, $\kappa$ increases by a factor 3 when the temperature rises by 400 K [37], reducing substantially the interference effect. In Fig. 3, we compute the absorption coefficient $A$ for silicon as a function of temperature and thickness, taking into account the temperature dependency $n$, $\kappa$ and $h$. Due to the thermally induced phase shift, the absorption $A$ of $\mu$m films can vary in large proportions with temperature. For $h = 1.2\,\mu$m for example, $A$ increases from a minimum of 18% at room temperature to a maximum of 63% at 460 °C. At elevated temperatures, $\kappa$ is high enough to kill the interference effects ($\alpha h \gg 1$), leading to an absorption of around 60% independent of temperature and film thickness.

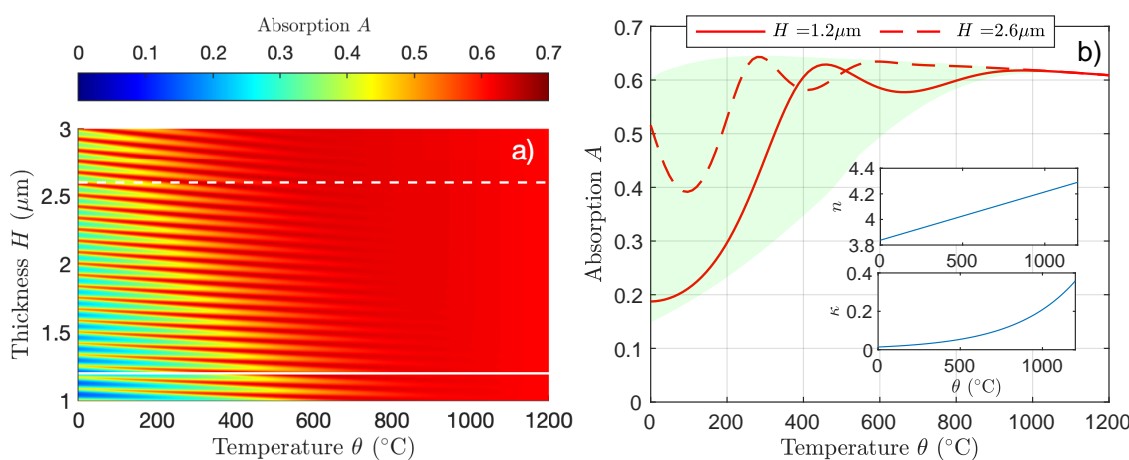

Figure 3:   a) Film absorption $A$ as a function of thickness and temperature at $\lambda = 641$ nm computed from Eqs. (3), using the optical properties for silicon displayed in the inset of b) from Ref. [37] (Eqs. (4)), and the thermal expansion coefficient of silicon from Ref. [40]). b) Absorption variations for $H = 1.2\,\mu$m and $H = 2.6\,\mu$m (cuts of top map along the 2 horizontal white lines). The amplitude and the rate of the absorption oscillations depend on the film thickness. The green area corresponds to the possible absorption values for film thicknesses in the range $1\,\mu$m $< H < 3\,\mu$m.

## 3.3   Laser induced cantilever heating: thermo-optical coupling

We consider a silicon cantilever of length $L$, width $W$, and thickness $H$ irradiated by a laser beam at its extremity (in $x = L$). The fraction $AP_0$ of light absorbed by the cantilever will result in a rise of its temperature. In vacuum, neglecting the thermal radiation, the only dissipation process is thermal conduction along the cantilever [17]. The stationary temperature profile can be estimated using the Fourier law

$$\frac{AP_0}{WH} = -k_{Si}(\theta(x))\frac{d\theta}{dx}, \tag{7}$$

where $k_{Si}$ is the thermal conductivity of silicon. The temperature dependency of $k_{Si}$ (displayed in the inset of Fig. 4) is significant and must be considered. Imposing the boundary condition of an isothermally clamped edge $\theta(x = 0) = \theta_0$, the temperature profile $\theta(x)$ solution of (7) reads as

$$\theta(x) = K_{Si}^{-1}\left(\frac{xAP_0}{WH}\right),\tag{8}$$

where $K_{Si}(\theta)$ is the primitive of $k_{Si}(\theta)$:

$$K_{Si}(\theta) = \int_{\theta_0}^{\theta} k_{Si}(\theta')d\theta'.\tag{9}$$

In Fig. 4, we report the temperature profiles $\theta(x)$ solving Eq. (8) for various absorbed power $AP_0$. Due to the significant diminution of silicon conductivity with temperature, the temperature profiles are clearly non-linear.

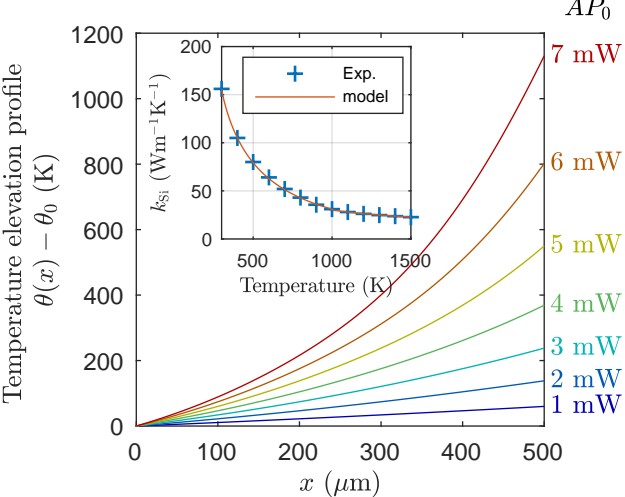

Figure 4: Theoretical temperature profiles taking into account the dependence of thermal conductivity of silicon on temperature for a cantilever with a cross-section $WH = 60\,\mu\text{m}^2$. The corresponding absorbed powers $AP_0$ are indicated on the right side. Inset: Temperature dependency of silicon thermal conductivity from Refs. [41,42].

For a relatively thick cantilever ($\alpha h \gg 1$), the interference effect is negligible, the absorption coefficient $A$ may be assumed constant. According to Eq. (8), the temperature variation with incident power $P_0$ at the fixed position $x = L$ will have the same behavior as the spatial profiles shown in Fig. 4 (i.e. using $P_0$ on the horizontal axis instead of $x$). For a cantilever such that $\alpha h \approx 1$, the absorption $A$ significantly depends on temperature. In that case, the absorbed power which allows to determine the temperature profile depends itself on the temperature $\theta_L = \theta(x = L)$ at the laser spot: there is a thermo-optical coupling. Eq. (8) implies that $\theta_L$ is a solution of

$$K_{Si}(\theta_L) = \frac{A(\theta_L, H, \lambda)LP_0}{WH}.\tag{10}$$

The temperature $\theta_L$ appears on both sides of (10). One can solve graphically (10) by comparing the curves $A(\theta_L, H, \lambda)$ and $WHK_{Si}(\theta_L)/LP_0$ (Fig. 5). At a given incident power $P_0$, the solutions correspond to the intersections between those two curves. For low or high power, there is a single solution. For intermediate powers, one finds three solutions. In that case, the

middle intersection corresponds to a situation where $\theta_L$ decreases when increasing $P_0$, this solution is unstable, i.e. the system is bistable. The corresponding solution curve $\theta_L$-vs-$P_0$ (inset of Fig. 5) presents an hysteresis; there is a range of forbidden temperature between 230 °C and 430 °C.

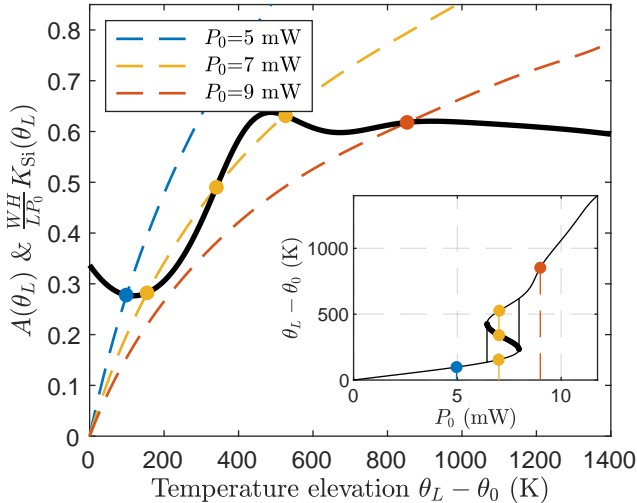

Figure 5: Graphical determination of the $\theta_L$-vs-$P_0$ curve for a silicon cantilever such that $H = 1.41\,\mu$m, $W = 27\,\mu$m and $L = 360\,\mu$m at $\lambda = 641$ nm. Depending on the incident power $P_0$ one finds 1 or 3 solutions. The intermediate intersection at $P_0 = 7$ mW is such that $\partial\theta_L/\partial P_0 < 0$ and thus corresponds to an unstable solution. Inset: $\theta_L$-vs-$P_0$ corresponding curve. The temperature presents an hysteresis. The thick line corresponds to the unstable branch of solutions. The temperature increases continuously up to a critical power of 8 mW where it jumps to the high temperature stable branch of the solution. For decreasing incident power, the temperature decreases continuously down to 6.4 mW where it drops to the low temperature stable branch.

The condition for the existence of bistability is that the slope of the curve $A(\theta_L)$ becomes higher than the slope of $WHK_{\mathrm{Si}}(\theta_L)/LP_0$ at an intersection. Note that this condition is more easily satisfied thanks to the strong non-linear rise of $\theta_L$ with $P_0$ predicted for a cantilever heated in vacuum. For a cantilever surrounded in air, the heat transfer through convection would tend to reduce this non-linearity and consequently make more difficult the observation of the bistability. In the case where the cantilever is in vacuum, the thickness $H$ is the only relevant geometric dimension for the apparition of the bistability. Indeed, according to Eq. (10), a change in width $W$ or length $L$ only modifies the powers involved in the characteristic curve but not its behavior. As we have seen in section 3.1, the absorption curve $A(\theta_L, H, \lambda)$ highly depends on thickness $H$ and the considered light wavelength $\lambda$. Thus, the characteristic curve $\theta_L$-vs-$P_0$ and the apparition of a potential hysteresis is also dependent on both parameters $H$ and $\lambda$. To illustrate these sensitivities, we display in Fig. 6 the incident power $P_0$ as a function of temperature elevation $\theta_L$ and thickness $H$ at both wavelengths 532 nm and 641 nm. The semi-transparent regions surrounded by the dashed lines correspond to the unstable solutions of Eq. (10). For any thickness displayed (1.25 $\mu$m $< H <$ 1.55 $\mu$m), the cantilever maximum temperature $\theta_L$ exhibits one or two bistable behaviors between 150 °C and 800 °C when irradiated at 641 nm while it remains continuous at 532 nm.

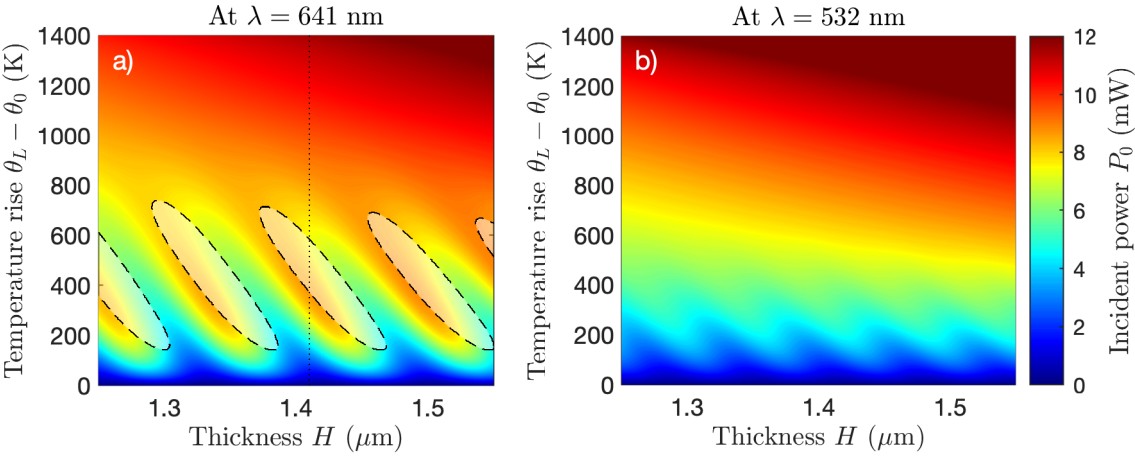

Figure 6: Incident power $P_0$ needed to reach a given temperature rise $\Delta\theta_L$ for a cantilever thickness $H$ solving Eq. (10) with $L = 360\,\mu$m and $W = 27\,\mu$m at 641 nm (a) and 532 nm (b). The dashed lines correspond to $\partial\theta_L/\partial P_0 = 0$ and surround unstable regions (semi transparent). Hystereses are only possible for $\lambda = 641$ nm, their position and size highly depend on $H$. The inset of Fig. 5 is a cut of this map along the vertical dotted line at $H = 1.41\,\mu$m.

## 4 Experimental results

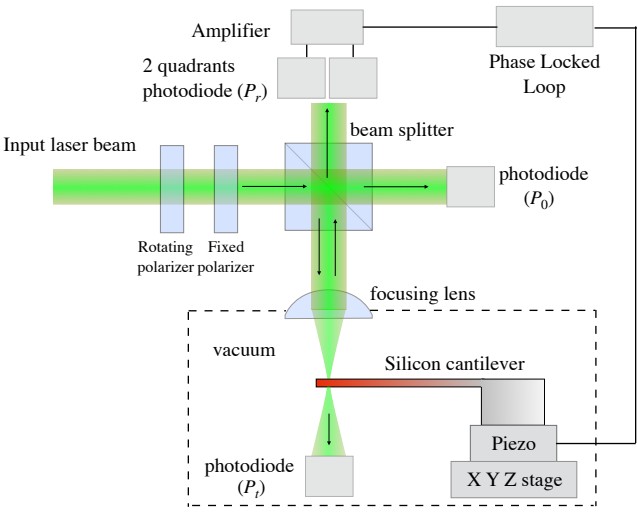

Figure 7: Sketch of the experimental setup. The silicon cantilever in vacuum is irradiated by a laser beam either at $\lambda = 532$ nm or $\lambda = 641$ nm focused at its free end. The laser allows to (i) heat the cantilever thanks to absorption, (ii) deduce the temperature rise by analyzing the frequency resonance shift thermally induced, (iii) measure optical coefficients $R$ and $T$.

The experiments presented in this section consist of heating a silicon cantilever in vacuum with a laser and measuring the evolution of its reflectivity $R$ and transmission $T$ along with the temperature $\theta_L$. When heated, the cantilever mechanical resonance frequencies shift owing to a change in the stiffness due to the temperature dependency of Young's modulus and thermal expansion. Knowing the thermo-mechanical properties of silicon, the cantilever temperature can be deduced by tracking the frequency shift [17]. The experimental setup uses a single laser beam (Fig. 7) to simultaneously heat the cantilever, measure the optical coefficients $R$ and $T$, and the temperature induced resonance frequency shift.

The laser beam, either from a solid state laser at $\lambda = 532\,$nm or from a laser diode at $\lambda = 641\,$nm, is focused with a lens at the cantilever free end. The beam waist at the cantilever surface has a radius of $5\,\mu$m. The incident beam intensity $P_0$ can be tuned continuously benefiting from light polarization, by rotating two linear polarizers relatively to each other. The reflected beam is centered on a two quadrants photodiode. The sum of the voltages delivered by the two quadrants measures the reflected intensity and allows to compute the reflectivity coefficient $R$. The voltage difference is sensitive to cantilever bending and is used as the input signal of a Phase Lock Loop (PLL, Nanonis OC4). The output signal of the PLL drives a piezo actuator which oscillates the cantilever at its tracked resonance. A photodiode placed under the cantilever measures the transmitted intensity and allows to compute the transmission coefficient $T$. One measurement consists in recording optical coefficients $R$, $T$ and tracking the resonance frequency as the incident power $P_0$ is continuously increased up to a maximal value then symmetrically decreased. The duration of one measurement is approximately 20 seconds. Since the characteristic time for heat diffusion is below $1\,$ms for a $500\,\mu$m long cantilever, the temperature profile can safely be considered stationary during all measurements. In the experiments presented herein, the cantilever is in a vacuum chamber at $10^{-2}\,$mBar and is heated from room temperature, $\theta_0 = 22\,°$C.

We perform the experiments on two different cantilevers: cantilever C14 is $L = 360\,\mu$m long, $W = 34\,\mu$m wide, and $H = 1.41\,\mu$m thick (MicroMasch HQ:CSC38), while cantilever C28 is $L = 513\,\mu$m long, $W = 31\,\mu$m wide and $H = 2.78\,\mu$m thick (Budget Sensors AIO-TL). These geometrical dimensions were measured using a scanning electron microscope (SEM) with uncertainties around $1\,$% for L and W, and $5\,$% for H. Both are uncoated tipless AFM silicon cantilevers. For each, we perform measurements using the two laser sources at $532\,$nm and $641\,$nm.

We first focus on measurements conducted with the relatively thin cantilever C14. The measured reflectivity $R$, transmission $T$ and deduced absorption[1] and $A = 1 - R - T$ when the cantilever is irradiated at 532 nm are plotted in Fig. 8-a. At low $P_0$, the cantilever is semi-transparent and exhibits relatively large variations ($\approx 35$%) in reflectivity $R$ and absorption $A$ due to the thermally induced cavity phase shift $\Delta\phi$ modulating the interferences. At high $P_0$, corresponding to high temperatures, the silicon extinction coefficient $\kappa$ is large enough to kill the interferences, the cantilever becomes non-transparent. The temperature elevation deduced from the frequency shift [17] measured for the first two mechanical modes is displayed in Fig. 8-b. We verify that the deduced temperature is independent of the resonance mode. We can observe the effect of the thermo-optical mechanism described in section 3.3: when absorption $A$ diminishes ($P_0 < 1.5\,$mW), the thermo-optical coupling tends to slow the temperature rise. Conversely, growing absorption ($1.5\,$mW $< P_0 < 5\,$mW) drives the temperature to rise faster. Note that the temperature data are displayed both for an increasing and decreasing power and cannot be distinguished between each other. This accurate superposition indicates that despite the thermo-optical coupling, the system remains stable and supports a single temperature at a given $P_0$. Up to an incident power of $13.6\,$mW, both optical coefficients and the temperature rise have excellent reproducibility, the cantilever undergoes only reversible physical changes. During the second mode measurement, an additional power of $1\,$mW was imposed leading to an irreversible phenomenon: the heating was strong enough to reach the melting point of silicon ($1410\,°$C) deteriorating the cantilever at the beam spot location.

Before damaging the cantilever, the same measurements were conducted using a different laser source at $\lambda = 641\,$nm (Fig. 8-c and 8-d). Owing to the weaker extinction coefficient $\kappa$ at higher wavelength, the interferences are more effective, leading to larger variations in

---

[1]Thanks to the small size of the laser spot, no light spills from the cantilever. Moreover, the very flat surfaces of silicon in AFM probes makes light scattering negligible.

absorption enhancing the thermo-optical coupling. In that case, the system can present multiple temperatures at a given $P_0$, exhibiting a large hysteresis around 7 mW and a smaller at 10.3 mW. For increasing power, the temperature jumps from 200 °C to 520 °C at $P_0 = 8.2$ mW. For decreasing power, the temperature drops from 410 °C to 130 °C at $P_0 = 6.8$ mW.

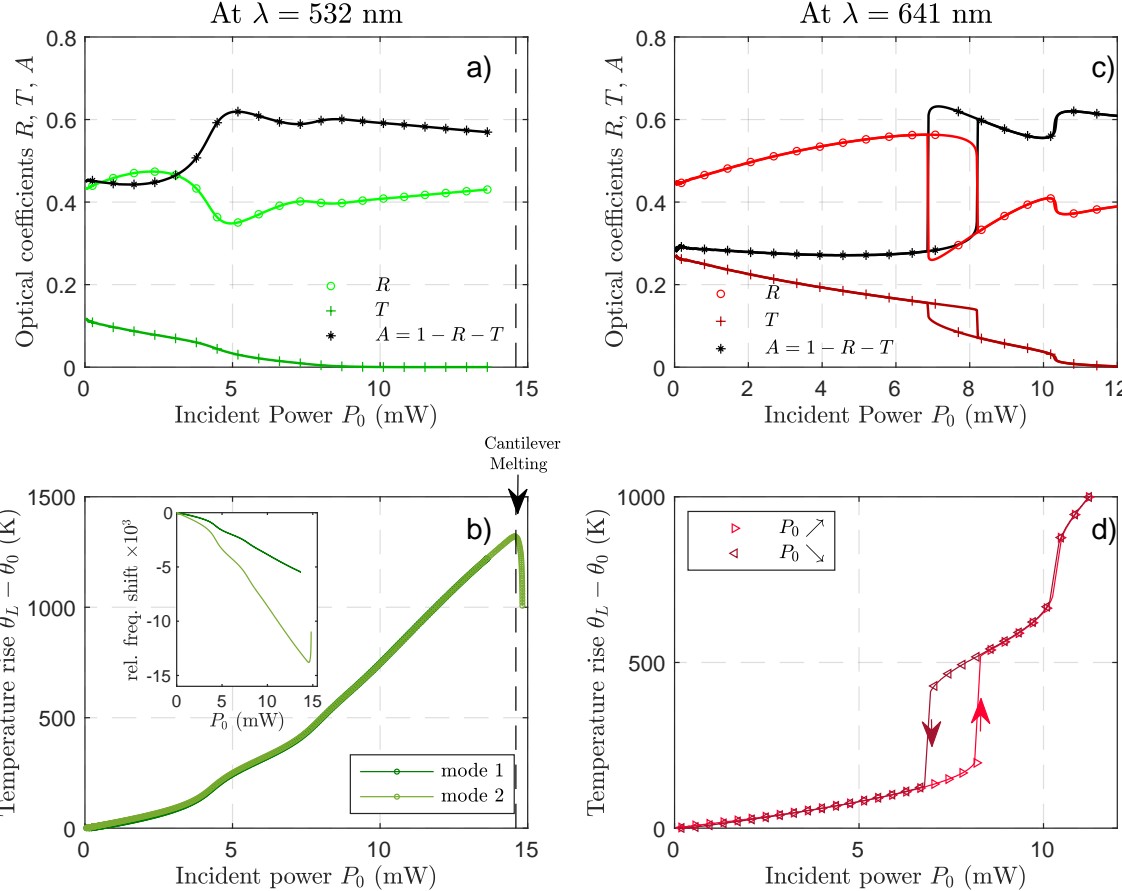

Figure 8: (Left) Cantilever C14 irradiated at $\lambda = 532$ nm: a) Reflectivity $R$, transmission $T$ and deduced absorption $A$ as a function of incident light power $P_0$. The observed variations are mainly induced by the change of silicon refractive index upon temperature rise. b) Temperature rise deduced from the frequency shift (for the first two modes). The temperature increases non regularly as a function of the incident power $P_0$. During the second mode shift measurement, at $P_0 = 14.6$ mW (vertical dashed line), the cantilever was molten at the laser waist. Inset: raw relative frequency shift for the first and second flexural modes. (Right) Cantilever C14 irradiated at $\lambda = 641$ nm: c) Reflectivity $R$, transmission $T$ and deduced absorption $A$ as a function of incident light power $P_0$. d) Temperature rise deduced from the first mode frequency shift. The temperature exhibits a large hysteresis between 6.8 mW and 8.2 mW.

In order to see the influence of thickness $H$, we perform similar measurements on the cantilever C28 (Fig. 9-b and -d). As for the cantilever C14, the characteristic $\theta_L$-vs-$P_0$ is a continuous function when the cantilever is irradiated at 532 nm while it becomes discontinuous exhibiting a small hysteresis (less than 0.1 mW wide) when irradiated at 641 nm. In that case, because of the thicker cantilever, the optical coefficient variations and the size of the induced hysteresis are smaller. Note that despite the factor two in thickness between C14 and C28, the level of involved power $P_0$ is comparable. Indeed for a given absorption power $AP_0$, the

temperature rise depends on the geometric factor $L/WH$ (see Eq.(10)), which only differs by 26% between the two cantilevers.

## 5   Quantitative comparison of theory vs experiment

In this section, we check that the measured variations in reflectivity $R$ and transmission $T$ can be quantitatively described by the Fabry-Pérot model presented in section 3. According to Eqs. (3), $R$ and $T$ at a given wavelength $\lambda$ are determined by the complex refractive index $\tilde{n} = n + i\kappa$ and the film thickness $h$. Thus, the measured reflectivity $R$ and transmission $T$ as a function of the temperature $\theta_L$ can be fully described, using Eqs. (4) and (5)[2]. Using this description, at each wavelength, we can perform a fit of $R$ and $T$ on the two cantilevers up to 500 °C, with $H_{14}$, $H_{28}$, $n_0$, $a_n$, $\kappa_0$, $\theta^\star$ as the fitting parameters, with $H_{14}$ and $H_{28}$ the thicknesses of cantilevers C14 and C28 respectively. We apply a least-squares method using simultaneously the datasets from the two cantilevers: a single set of optical parameters is deduced and should describe both thicknesses.

The best fit, reported in Fig. 9, is obtained for thicknesses $H$ only 2% different from the SEM measurements. For temperatures up to 500 °C, all experimental curves are in remarkable agreement with the obtained fit. The silicon optical parameters deduced from the fit are listed in table 1 along with values from literature [37].

| refractive index coeff. | | Jellison [37] | our fit | diff. |
|:---:|:---:|:---:|:---:|:---:|
| | $n_0$ | 4.115 | 4.121 | 0.2 % |
| $\lambda = 532\,\mathrm{nm}$ | $a_n\ (10^{-4}\,\mathrm{K}^{-1})$ | 1.213 | 1.154 | -4.9 % |
| | $\kappa_0$ | 0.0326 | 0.0333 | 2.1 % |
| | $\theta^\star\ (°C)$ | 369 | 344 | -6.8 % |
| | $n_0$ | 3.840 | 3.838 | -0.04% |
| $\lambda = 641\,\mathrm{nm}$ | $a_n\ (10^{-5}\,\mathrm{K}^{-1})$ | 9.846 | 9.832 | -0.1 % |
| | $\kappa_0$ | 0.0141 | 0.0162 | 14.6% |
| | $\theta^\star\ (°C)$ | 370 | 364 | -1.6 % |

Table 1: Optical coefficients of the complex refractive using index (using eqns (4)) at both wavelength 532 nm and 641 nm deduced from the fit of $T$ and $R$ measured with both cantilevers C14 and C28 displayed Fig. 9. The obtained values are in good agreement with silicon values measured by Jellison et al [37]. Note that $\kappa_0$ are given here for $\theta$ expressed in °C in Eq. 4b to ease comparison with Ref. [37].

Although the measurement of silicon complex refractive index was not the aim of this work, we retrieve accurately its values and its temperature dependence. For temperature above 500 °C, we plot in Fig. 9 the theoretical coefficients $R$ and $T$ using the extrapolated refractive index deduced from the fit. In this range of temperatures, the measured coefficients $R$ and $T$ at 641 nm with C14 slightly deviates from the extrapolation suggesting that the simple model of Eqs.(4) becomes inadequate to describe the variation of silicon refractive index at high temperatures.

---

[2]It must be noted that due to the finite size of the laser waist, the transmitted and reflected intensity probe the local cantilever properties under the beam spot. Therefore, the use of Eqs. (3) at a unique temperature implicitly assumes a uniform temperature under the beam. To justify this assumption, we performed a 2D simulation computing the temperature distribution of a cantilever heated by a gaussian beam with the measured radius of 5 $\mu$m. For both cantilever geometries (C14 and C28), the standard deviation of the temperature distribution weighted by the gaussian beam is found to be only 1.5 % of the mean temperature. Such a low temperature dispersion allows us to describe the measured optical coefficients $R$ and $T$ with a unique temperature.

Using the complex refractive index deduced from the fit presented above, we can compute the theoretical characteristic $\theta_L$-vs-$P_0$ solving Eq. (10). In Fig. 9, we compare the theoretical predictions with experimental curves for both cantilevers irradiated at both wavelengths. In order to make the predictions coincide with experiments, the thermal conductivity used to compute the function $K_{Si}$ for cantilevers C14 and C28 was respectively chosen 18% and 15% lower than bulk silicon values[3] [17, 43, 44]. The thermo-optical model exposed in section 3.3 accurately predicts the hysteresis observed with both cantilevers.

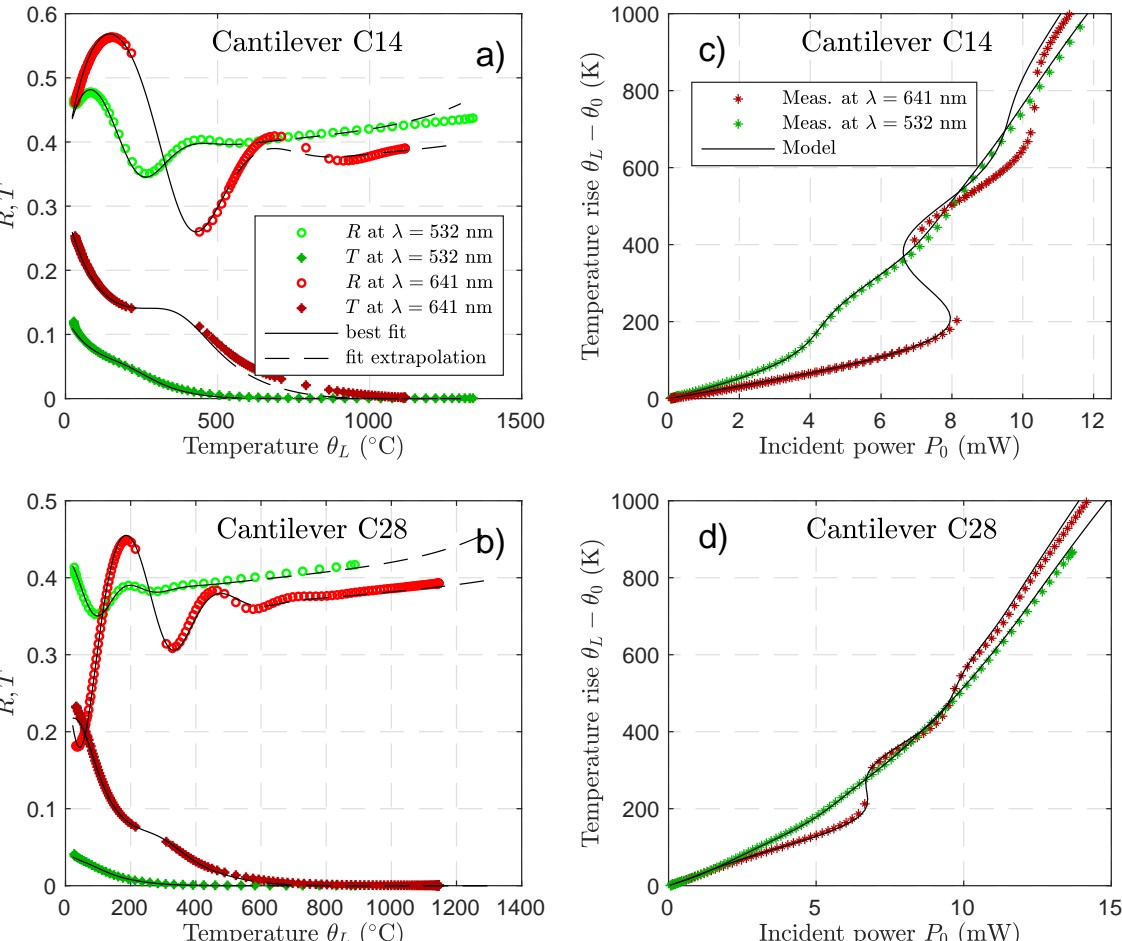

Figure 9: (left) Temperature variation of $R$ and $T$ measured for cantilever C14 (a) and C28 (b) illuminated at both wavelengths 532 nm and 641 nm. The black curves correspond to the fit of Eqs. (3) with the complex refractive index given by Eqs. (4) and the thickness given by Eq. (5) with $H = H_{14} = 1435$ nm for C14 and $H = H_{28} = 2740$ nm for C28. At temperatures above 500 °C, the black dashed curves correspond to the extrapolation of the model with the coefficients given by the fit. The optical coefficients $n_0$, $a_n$, $\kappa_0$ and $\theta^\star$ resulting from the fit are listed in table 1. (right) Temperature rise as a function of incident power measured for cantilever C14 (c) and C28 (d) at both wavelengths, compared with theoretical predictions solving Eq. (10) using the complex refractive index deduced from the fit displayed on the left. Our model recovers the various characteristics $\theta_L$-vs-$P_0$ observed in four configurations and accurately predicts the bistability appearances.

---

[3]This reduced thermal conductivity can be explained by the effect of phonon scattering at interfaces occurring at micrometer scale. It is actually temperature dependent, so that the function $k_{Si}(\theta)$ should be modified. However, in lack of quantitative data of this effect, and since only minute changes to the estimation of the temperature rise at the laser spot position are expected, we only use a global coefficient.

# 6 Conclusion

We describe in this article a surprisingly simple system presenting optical bi-stability: a simple raw silicon cantilever, with a visible laser beam incident perpendicularly at its free end. We demonstrate that the cantilever thickness acts as a lossy Fabry-Pérot cavity, prompting non-monotonous absorption of light as temperature $\theta$ is changed. Since temperature and absorption are coupled by heat conduction, it results in a non-linear optical system that can present multi-stability. The mechanism described here is embedded in the previous approaches studying thermo-optical bistability [22, 25, 31, 32]. It it however noticeable that silicon plays in our experiment the role of both the absorber and the temperature sensitive refractive material: there is no need for external mirrors or coatings to observe the effect. Thanks to the wide knowledge of physical properties of silicon, the model describes quantitatively the experimental observations on a large temperature range with no adjustable parameters. It allows retrieving the evolution of the complex refractive index of silicon from ambient to high temperature.

Obviously, the optical power in the mW range and the rather slow switching time in the ms range (thermal diffusion in silicon over $500\,\mu$m) don't match with the ideal requirements of all-optical signal processing components. Devices with $5\,$fJ operating energy and $20\,$ps response times have indeed been demonstrated [26]. However, some optimisation of our current samples (basically AFM cantilevers) could certainly be performed to reduce both optical powers and response time of the system. Indeed, from Eq. (10) and its graphical resolution of Fig. 5, one can see that for a given thickness $H$, the bistability is driven by the temperature, itself driven by the product of the input light power $P_0$ and the thermal resistance $\sim L/(WHk_{\mathrm{Si}})$. The minimum power to trigger the instability could in principle be arbitrary low as one could tune for example $W$ to very low values. The switching time could be optimised on its own by shortening the device. The interplay between geometry and fast/low power operation would then lead to a racket like shape design, with an area large enough to accommodate the laser on the Fabry-Pérot cavity side, and a minimalist short leg of high thermal resistance connecting the optical cavity to the thermalised support. Though it still wouldn't be competitive with state-of-the-art components, optimisation of such a device is at reach with the ingredients of our current work. But beyond this application area, we believe our work brings several interesting perspectives in other domains.

First, for AFM sensors, it provides a framework to understand why reflectivity and transmission of cantilevers can vary in a large range with subtle geometry changes. Coated cantilever should be less sensitive to these effects, though usual coating thicknesses (a few tens of nm) are often not large enough to kill all light transmission and thus to prevent any interference in the cantilever bulk. Even if usual laser powers in standard AFM imaging conditions are too low to trigger the bistability effect, keeping in mind the thermo-optical coupling could be useful to discard possible artifacts from experiments.

Second, for light actuated cantilevers by thermo-mechanical coupling [5–8], the large temperature change one can trigger with limited power excursion in the hysteretic area could be beneficial. One could reach very large actuation amplitudes using the non-monotonous response. This would require engineering the thermal time response (i.e. laser spot distance to the cantilever base) to be faster than the resonance period of the cantilever.

Last, the Fabry-Pérot effect could be used to reach huge sensitivity to thermal fluxes. Indeed, with adequate thickness and incident power, the behavior may be prepared just tangent to the hysteretic one (close to the behavior of cantilever C14 around $10\,$mW in fig. 9 for example). In this case, we have $\partial\theta_L/\partial P_0 \sim \infty$: the temperature of the cantilever end is extremely sensitive to any heat flux variation. Heat exchanges between the tip and the sample in thermal AFM [45], for example change slightly $P_0$. They would have a huge thermal effect that could

be sensed by the optical (e.g. reflectivity) or mechanical properties. It should be mentioned however that other noises in the experiment (such as laser intensity fluctuations) would also be amplified by the large sensitivity. In a different field of application, single molecule light absorption imaging [2] could benefit from engineered resonators to enhance sensitivity.

# Acknowledgement

We thank Artyom Petrosyan and Sergio Ciliberto for enlightening technical and scientific discussions.

**Funding information** Part of this research has been funded by the ANR projects HiResAFM (ANR-11-JS04-012-01) and STATE (ANR-18-CE30-0013) of the Agence Nationale de la Recherche in France.

**Data availability** The data that support the findings of this study are openly available in Zenodo at https://doi.org/10.5281/zenodo.4703793 [46].

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
