# Peer review of "Thermo-optical bistability in silicon micro-cantilevers"

_SciPost Physics_

## Round 1 · Referee Report · Anonymous (Referee 1) · 2021-3-8

Report

The authors report on a thermo-mechanical bi-stability in a microfabricated cantilever. The effect results in a large variation of the transmitted and reflected laser power. The bi-stability is described in an accurate way by a model based on the temperature dependence of the refractive index and the thermal expansion of the cantilever. The manuscript is written in a precise way. I recommend publication.
  • validity: high
  • significance: good
  • originality: good
  • clarity: high
  • formatting: good
  • grammar: good

Author:  Ludovic Bellon  on 2021-04-20  [id 1372]

(in reply to Report 1 on 2021-03-08)

We thank the reviewer for his positive review of our manuscript.

---

## Round 1 · Referee Report · Anonymous (Referee 2) · 2021-3-17

Strengths

1- Excellent combination of theoretical model and experimental confirmation of developed thermo-optical bistability in silicon cantilevers 2- Clear structure 3- Well written and understandable 4- Good use of visualisations

Weaknesses

1- Potentially too little emphasis on possible applications of the observed bistability regarding the necessary high optical powers

Report

In their manuscript titled “Thermo-optical bistability in silicon micro-cantilevers”, the authors discuss the occurrence of thermo-optic bistability in a fundamental single-material system of a micron-sized silicon cantilever. They construct a simple theoretical model to combine the modulation of optical absorption due to interference in Fabry-Pérot cavities with the temperature-dependent changes of the complex refractive index of the material, resulting in a parameter range of incident laser power that allows for bistable temperature operation.
The article is very well written, with a nice read thread following through the whole text. I specifically also want to commend the authors for their clear language and well selected visualizations. During the reading of the manuscript, some questions that were triggered by certain text blocks were instantly answered in the following section, highlighting the clear message of the authors.

In general, I only have minor comments to the authors, which go mostly towards comparing their results with existing concepts of optical bistability in silicon microring resonators, but might lead away from the currently well thought out and concise storyline:

• The only point where I would have hoped for a slightly more in-depth explanation was for the connection between the positive/negative thermo-optic coupling of the system and the positive feedback/stability regarding temperature variations due to absorption. I think the authors look here in section II at their full system, but if one conceptually thinks about on which side of a Fabry-Pérot resonance for a given cantilever geometry the excitation wavelength will be, shouldn’t there be both, positive feedback and stability possible by just tuning the excitation wavelength to the other side of the resonance? Or is this exactly what the authors want to state here?
• While their best fit in Figure 9 is indeed in excellent agreement with their experimental data, I was wondering how the used thicknesses H were determined? In essence, I can’t fully follow the authors how they ended up at a 2% difference of the chosen value with respect to SEM measurements. This variation is indeed fully within the measurement accuracy of the SEM images, I was only wondering how this adaption was done.
• For the choice of using a 18%/15% lower thermal conductivity than bulk silicon, I was wondering if this was a constant, temperature-independent change, or if a different thermal response curve would fit the data equally well?
• It would be interesting to state the Q factor of the cantilever resonance and the minimum threshold power to observe the bistable operation. This would put the work in context to thermo-optic bistability in microring resonators. While the authors clearly state that the investigated system is not designed for its fast switching times or utilization as optical memory (and thus a comparison with these specifically designed systems is not really fair), it could allow readers from that background to more easily understand the system’s operation point.
• One question that somehow remained unanswered was what the optimum cantilever height for one given wavelength is so that thermo-optic bistability occurs at minimum input powers.
• At the beginning of section II, the authors state that due to the poor reflectivity of silicon of only 37%, the material thus absorbs a significant fraction of visible light. I am not sure if I would really combine these two aspects without stating the bandgap position of silicon. For example, AlAs exhibits only significant absorption below 570nm, while still retaining a high refractive index of 3.1, meaning a reflectivity of also around 26%. I agree with the authors that due to Kramers-Kronig relations, the high refractive index is linked to a wide band of absorption, but this could be shifted outside of the spectral region of interest. This is just nit-picking here, I don’t think a real change is required, maybe just a small reformulation of the statement.

With these very minor points in mind, I can nonetheless fully recommend publication in SciPost Physics.

Requested changes

1- In the figure caption of Fig. 3, it should read “green area” instead of “grey area” 2- On page 5 before equation (10), I think “is a solution of” is correct 3- In the figure caption of Fig. 8, it should read “vertical dashed line”, and in Fig. 9 “black dashed curves correspond”

  • validity: top
  • significance: good
  • originality: high
  • clarity: top
  • formatting: excellent
  • grammar: excellent

Author:  Ludovic Bellon  on 2021-04-20  [id 1373]

(in reply to Report 2 on 2021-03-17)
Category:
answer to question

We thank the reviewer for his very positive review of our manuscript. All the minor comments mentioned by the reviewer were answered.

The only point where I would have hoped for a slightly more in-depth explanation was for the connection between the positive/negative thermo-optic coupling of the system and the positive feedback/stability regarding temperature variations due to absorption. I think the authors look here in section II at their full system, but if one conceptually thinks about on which side of a Fabry-Pérot resonance for a given cantilever geometry the excitation wavelength will be, shouldn’t there be both, positive feedback and stability possible by just tuning the excitation wavelength to the other side of the resonance? Or is this exactly what the authors want to state here?

Exactly: the sign of the thermo-optical coupling feedback and the side on which we stand with respect to the Fabry-Pérot resonance are directly linked. Indeed, the interferences inside the cavity are fully determined by the optical phase, given by equation (6). We see that increasing the temperature $\theta$, the thickness $H$ or decreasing the wavelength $\lambda$ have the same effect when exploring the resonance. A complication when tuning $\lambda$ is that coefficients $n_0$ and $a_n$ also depend on $\lambda$, making things slightly more complicated than changing $H$ or $\theta$. If we conceptually forget about this dependence anyway, then when $\lambda$ is above the resonance, decreasing $\lambda$ or increasing $\theta$, one gets closer to the resonance thus increases the absorption $A$: this is a destabilising feedback ($A$ increases with $\theta$). By symmetry, on the other side of the resonance the feedback is stabilising ($A$ decreases with $\theta$). A new paragraph has been added after eq. (6) for this deeper explanation of the connection between the resonance and the thermo-optical coupling.

While their best fit in Figure 9 is indeed in excellent agreement with their experimental data, I was wondering how the used thicknesses $H$ were determined? In essence, I can’t fully follow the authors how they ended up at a 2% difference of the chosen value with respect to SEM measurements. This variation is indeed fully within the measurement accuracy of the SEM images, I was only wondering how this adaption was done.

The thicknesses $H$ ($H_{14}$ and $H_{28}$ for cantilever C14 and C28) are determined from the fit, which was indeed not obvious in our initial manuscript. We corrected this issue in the new version. Because the model describing the optical coefficients (reflectivity and transmission) consists of functions oscillating with the phase change $\phi=4\pi n h/\lambda$, in practice the sum of squared residuals presents many local minima and the fitting procedure does not converge systematically towards the optimal set of parameters $H_{14}$, $H_{28}$, $n_0$, $a_n$, $\kappa_0$, $\theta^\star$ (the global minimum) starting from any initial guess value for the thicknesses $H_{14}$ and $H_{28}$ and the real refractive index $n_0$. For this reason, the fit is not performed blindly by the minimisation of the residual with 6 free parameters, but we perform many fits with fixed thicknesses in a 2D map around the SEM values $\pm{200}{nm}$, and with several initial guess for $n_0$, to be sure to find the optimal set of parameters. The optimal thicknesses of the global residual minima is only 2% different from the SEM measurements.

For the choice of using a 18%/15% lower thermal conductivity than bulk silicon, I was wondering if this was a constant, temperature-independent change, or if a different thermal response curve would fit the data equally well?

Because the spectra of phonon mean free path shifts to shorter lengths with increasing temperature, the confinement effect is expected to decrease with temperature. According to the literature [A. S. Henry and G. Chen, Journal of Computational and Theoretical Nanoscience 5, 141 (2008)], the contribution in thermal conductivity of phonon having mean free path superior to a few micrometers typically decreases from 20% at 300K to 12% at 1000K. Thus, the correction factor of 18%/15% taking into account the phonon confinement effect should rigorously be temperature-dependent as suggested by the referee. For the sake of simplicity, we did not take into this temperature dependence in our article. This simplification has however very little influence in determining the correct temperature rise at the laser spot position [B. Pottier, F. Aguilar Sandoval, M. Geitner, F. Melo, and L. Bellon, Resonance frequency shift of silicon cantilevers heated from 300K up to the melting point, arXiv:2012.00421 (2020)]. We added a footnote to the manuscript to specify this detail.

It would be interesting to state the Q factor of the cantilever resonance and the minimum threshold power to observe the bistable operation. This would put the work in context to thermo-optic bistability in microring resonators. While the authors clearly state that the investigated system is not designed for its fast switching times or utilization as optical memory (and thus a comparison with these specifically designed systems is not really fair), it could allow readers from that background to more easily understand the system’s operation point.

We added a paragraph at the end of section 3.1 to describe the low quality of the optical resonator: The silicon film thus acts as a lossy Fabry-Pérot cavity. Due to the high absorption, it is an intrinsically bad resonator: at $\lambda=641$ nm for $h=2$ µm, the wavelength full-width at half-maximum of the transmission oscillation amplitude is $\delta \lambda \sim 16$ nm, while the free spectral range is $\Delta \lambda \sim 21$ nm, leading to a finesse $F = \Delta \lambda / \delta \lambda \sim 1.3$, and a quality factor of the optical resonator $Q = \lambda / \delta \lambda \sim 40$. Better optical figures of merit could be reach by tuning the wavelength towards infrared.

We address the minimum power in the next point.

One question that somehow remained unanswered was what the optimum cantilever height for one given wavelength is so that thermo-optic bistability occurs at minimum input powers.

To answer this question, let us refer to Eq. 10 and its graphical resolution of Fig. 5. One can see that for a given thickness $H$, the bistability is driven by the temperature, itself driven by the product of the input light power $P_0$ and the thermal resistance $\sim L/(W H k_{Si})$. The minimum power could in principle be arbitrary low as one could tune for example the width of the cantilver to very low values. $L$ could be increased as well to lower the threshold in power, but at the expense of an increased thermal relaxation time. Obviously, at some point thermal radiation would take over conduction for heat dissipation, but this is beyond the current description. In Fig. 6, the minimum power to trigger the bistability can be read from the color scale for the geometry described in the caption, and is close to 4 mW at the lower right extremity of the unstable areas. Exploring an extended thickness range reveals that this is near to the optimum (it gets a bit lower when lowering $H$, but the hysteresis gets stronger, up to melting the cantilever directly at the first instable point below 1 µm). This 4 mW minimum power is directly translated into 0.4 mW for $W=2.7$ µm for example. An even lower threshold power could be attained by further increasing the thermal resistance, engineering the shape of the beam away from a rectangular cantilever: one could for example design a racket like shape, with an area large enough to accommodate the laser on the Fabry-Pérot cavity side, and a minimalist leg of high thermal resistance connecting the optical cavity to the thermalised support. Optimisation of such a device is at reach with the ingredients of our current work, but beyond its scope ! We nevertheless added a few ideas about this discussion in the conclusion of the manuscript.

At the beginning of section II, the authors state that due to the poor reflectivity of silicon of only 37%, the material thus absorbs a significant fraction of visible light. I am not sure if I would really combine these two aspects without stating the bandgap position of silicon. For example, AlAs exhibits only significant absorption below 570 nm, while still retaining a high refractive index of 3.1, meaning a reflectivity of also around 26%. I agree with the authors that due to Kramers-Kronig relations, the high refractive index is linked to a wide band of absorption, but this could be shifted outside of the spectral region of interest. This is just nit-picking here, I don’t think a real change is required, maybe just a small reformulation of the statement.

We reformulated the approximative sentence: For visible light, silicon is neither transparent nor a good mirror (reflectivity of around 37%).

Weaknesses: Potentially too little emphasis on possible applications of the observed bistability regarding the necessary high optical powers

The discussion above on the minimum optical powers, and its inclusion in the conclusion of the manuscript hopefully addresses the light weakness noted here by the referee.

Requested changes...

All three requested changes were taken into account in the new manuscript.

---

## Round 1 · Referee Report · Anonymous (Referee 3) · 2021-3-24

Strengths

  1. The work is a thorough study of thermo-optic bistability in silicon cantilevers, which are widely applied and basic systems.
  2. There is excellent support of experimental findings with theory.
  3. The findings are interesting, especially given the high optical loss of these silicon 'resonators'
  4. The paper is very well written and comprehensive.

Weaknesses

I have very little weaknesses to report. A minor comment is that 1. even though cantilevers are widely employed in e.g. force microscopy, it is not specified concretely which, if any, current applications would immediately benefit from the understanding gained in this work: at least, there are no citations to works that pose outstanding questions that are answered by this work. This is however not necessary, and it may very well be that such connections will be recognized later.

Report

I read this manuscript with pleasure. I believe the criteria for publication are met; see my strengths and weaknesses assessment.

Requested changes

At the very end of the paper, the authors speculate about using the bistability for sensing with 'infinite sensitivity'. Here, it would be good to acknowledge that the usefullness of bistabilities to enhance practical sensing applications is not as clear-cut as they make it seem: At a bistability, the sharp threshold for a small parameter change is also greatly amplifying any noise mechanism. As such, it is not self-evident that the bistability would make it easier to discern a signal on top of noise - as any useful sensor should aim to do. I recommend that the authors discuss this aspect in a more nuanced fashion.

  • validity: top
  • significance: good
  • originality: high
  • clarity: top
  • formatting: excellent
  • grammar: excellent

Author:  Ludovic Bellon  on 2021-04-20  [id 1374]

(in reply to Report 3 on 2021-03-24)
Category:
answer to question

We thank the reviewer for his very positive review of our manuscript.

Weaknesses: I have very little weaknesses to report. A minor comment is that even though cantilevers are widely employed in e.g. force microscopy, it is not specified concretely which, if any, current applications would immediately benefit from the understanding gained in this work: at least, there are no citations to works that pose outstanding questions that are answered by this work. This is however not necessary, and it may very well be that such connections will be recognized later.

This work is indeed curiosity driven: we observe, understand and fully characterise an unexpected and however striking effect in a very simple system. We are not aware of research tracks where the understanding gained in our paper would directly answer current hot questions. In the conclusion, we do mention however some interesting leads and perspectives, and believe the wide audience of SciPost Physics could trigger connections and links with more applications. Moreover, in light of referee 2's comments, we strengthen the conclusions with possible engineering optimisation leads to address the "slow" and "high power" issues of the current realisation.

Requested changes: At the very end of the paper, the authors speculate about using the bistability for sensing with 'infinite sensitivity'. Here, it would be good to acknowledge that the usefullness of bistabilities to enhance practical sensing applications is not as clear-cut as they make it seem: At a bistability, the sharp threshold for a small parameter change is also greatly amplifying any noise mechanism. As such, it is not self-evident that the bistability would make it easier to discern a signal on top of noise - as any useful sensor should aim to do. I recommend that the authors discuss this aspect in a more nuanced fashion.

We modified the paragraph to take into account the wise referee's concern, nuancing our discussion. Note that we suggest to get at the brink of the bistable region, to benefit from the increased sensitivity without dealing with the complexity of a detection with an hysteresis.

---

## Editorial Decision

resubmitted